# LEARNED THRESHOLD PRUNING

## ABSTRACT

This paper presents a novel differentiable method for unstructured weight pruning of deep neural networks. Our learned-threshold pruning (LTP) method learns per-layer thresholds via gradient descent, unlike conventional methods where they are set as input. Making thresholds trainable also makes LTP computationally efficient, hence scalable to deeper networks. For example, it takes 30 epochs for LTP to prune ResNet50 on ImageNet by a factor of 9.1. This is in contrast to other methods that search for per-layer thresholds via a computationally intensive iterative pruning and fine-tuning process. Additionally, with a novel differentiable $L_0$ regularization, LTP is able to operate effectively on architectures with batch-normalization. This is important since $L_1$ and $L_2$ penalties lose their regularizing effect in networks with batch-normalization. Finally, LTP generates a trail of progressively sparser networks from which the desired pruned network can be picked based on sparsity and performance requirements. These features allow LTP to achieve competitive compression rates on ImageNet networks such as AlexNet ($26.4\times$ compression with $79.1\%$ Top-5 accuracy) and ResNet50 ($9.1\times$ compression with $92.0\%$ Top-5 accuracy). We also show that LTP effectively prunes modern *compact* architectures, such as EfficientNet, MobileNetV2 and MixNet.

## 1 INTRODUCTION

Deep neural networks (DNNs) have provided state-of-the-art solutions for several challenging tasks in many domains such as computer vision, natural language understanding, and speech processing. With the increasing demand for deploying DNNs on resource-constrained edge devices, it has become even more critical to reduce the memory footprint of neural networks and also to achieve power-efficient inference on these devices. Many methods in model compression Hassibi et al. (1993); LeCun et al. (1989); Han et al. (2015b); Zhang et al. (2018), model quantization Jacob et al. (2018); Lin et al. (2016); Zhou et al. (2017); Faraone et al. (2018) and neural architecture search Sandler et al. (2018); Tan & Le (2019a); Cai et al. (2018); Wu et al. (2019) have been introduced with these goals in mind.

Neural network compression mainly falls into two categories: structured and unstructured pruning. Structured pruning methods, e.g., He et al. (2017); Li et al. (2017); Zhang et al. (2016); He et al. (2018), change the network's architecture by removing input channels from convolutional layers or by applying tensor decomposition to the layer weight matrices whereas unstructured pruning methods such as Han et al. (2015b); Frankle & Carbin (2019); Zhang et al. (2018) rely on removing individual weights from the neural network. Although unstructured pruning methods achieve much higher weight sparsity ratio than structured pruning, unstructured is thought to be less hardware friendly because the irregular sparsity is often difficult to exploit for efficient computation Anwar et al. (2017). However, recent advances in AI accelerator design Ignatov et al. (2018) have targeted support for highly efficient sparse matrix multiply-and-accumulate operations. Because of this, it is getting increasingly important to develop state-of-the-art algorithms for unstructured pruning.

Most unstructured weight pruning methods are based on the assumption that smaller weights do not contribute as much to the model's performance. These pruning methods iteratively prune the weights that are smaller than a certain threshold and retrain the network to regain the performance lost during pruning. A key challenge in unstructured pruning is to find an optimal setting for these pruning thresholds. Merely setting the same threshold for all layers may not be appropriate because the distribution and ranges of the weights in each layer can be very different. Also, different layers may have varying sensitivities to pruning, depending on their position in the network (initial layers versus final layers) or their type (depth-wise separable versus standard convolutional layers). The

best setting of thresholds should consider these layer-wise characteristics. Many methods Zhang et al. (2018); Ye et al. (2019); Manessi et al. (2018) propose a way to search these layer-wise thresholds but become quite computationally expensive for networks with a large number of layers, such as ResNet50 or EfficientNet.

In this paper, we propose Learned Threshold Pruning (LTP) to address these challenges. Our proposed method uses separate pruning thresholds for every layer. We make the layer-wise thresholds trainable, allowing the training procedure to find optimal thresholds alongside the layer weights during finetuning. An added benefit of making these thresholds trainable is that it makes LTP fast, and the method converges quickly compared to other iterative methods such as Zhang et al. (2018); Ye et al. (2019). LTP also achieves high compression on newer networks Tan & Le (2019a); Sandler et al. (2018); Tan & Le (2019b) with squeeze-excite Hu et al. (2018) and depth-wise convolutional layers Chollet (2017).

Our key contributions in this work are the following:

- We propose a gradient-based algorithm for unstructured pruning, that introduces a learnable threshold parameter for every layer. This threshold is trained jointly with the layer weights. We use soft-pruning and soft $L_0$ regularization to make this process end-to-end trainable.
- We show that making layer-wise thresholds trainable makes LTP computationally very efficient compared to other methods that search for per-layer thresholds via an iterative pruning and finetuning process, e.g., LTP pruned ResNet50 to 9.11x in just 18 epochs with 12 additional epochs of fine-tuning, and MixNet-S to 2x in 17 epochs without need for further finetuning.
- We demonstrate state-of-the-art compression ratios on newer architectures, i.e., $1.33\times$, $3\times$ and $2\times$ for MobileNetV2, EfficientNet-B0 and MixNet-S, respectively, which are already optimized for efficient inference, with less than $1\%$ drop in Top-1 accuracy.
- The proposed method provides a trace of checkpoints with varying pruning ratios and accuracies. Because of this, the user can choose any desired checkpoint based on the sparsity and performance requirements for the desired application.

## 2 RELATED WORK

Several methods have been proposed for both structured and unstructured pruning of deep networks. Methods like He et al. (2017); Li et al. (2017) use layer-wise statistics and data to remove input channels from convolutional layers. Other methods apply tensor decompositions on neural network layers, Denton et al. (2014); Jaderberg et al. (2014); Zhang et al. (2016) apply SVD to decompose weight matrices and Kim et al. (2015); Lebedev et al. (2014) apply tucker and cp-decompositions to compress. An overview of these methods can be found in Kuzmin et al. (2019). These methods are all applied after training a network and need fine-tuning afterwards. Other structured methods change the shape of a neural network while training. Methods like Bayesian Compression Louizos et al. (2017), VIBnets Dai et al. (2018) and L1/L0-regularization Srinivas et al. (2017); Louizos et al. (2018) add trainable gates to each layer to prune while training.

In this paper we consider unstructured pruning, i.e. removing individual weights from a network. This type of pruning was already in use in 1989 in the optimal brain damage LeCun et al. (1989) and optimal brain surgeon Hassibi et al. (1993) papers, which removed individual weights in neural networks by use of Hessian information. More recently, Han et al. (2015a) used the method from Han et al. (2015b) as part of their full model compression pipeline, removing weights with small magnitudes and fine-tuning afterwards. This type of method is frequently used for pruning, and has recently been picked up for finding DNN subnetworks that work just as well as their mother network in Frankle & Carbin (2019); Zhou et al. (2019). Another recent application of Han et al. (2015b) is by Renda et al. (2020) where weight and learning-rate rewinding schemes are used to achieve competitive pruning performances. These methods, however, are very computationally extensive requiring many hundreds of epochs of re-training. Finally, papers such as Molchanov et al. (2017); Ullrich et al. (2017) apply a variational Bayesian framework on network pruning.

Other methods that are similar to our work are Zhang et al. (2018) and Ye et al. (2019). These papers apply the alternating method of Lagrange multipliers to pruning, which slowly coaxes a network into pruning weights with a L2-regularization-like term. One problem of these methods is that they are

time-intensive, another is that they need manual tweaking of compression rates for each layer. In our method, we get rid of these restrictions and achieve comparable compression results, at fraction of the computational burden and without any need for setting per-layer pruning ratios manually. Kusupati et al. (2020) and Manessi et al. (2018) learn per-layer thresholds automatically using soft thresholding operator or a close variant of it. However they rely on $L_1$ and/or $L_2$ regularization, which as shown in section 3.2, is inefficient when used in networks with batch-normalization Ioffe & Szegedy (2015). He et al. (2018) use reinforcement learning to set layer-wise prune ratios for structured pruning, whereas we learn the pruning thresholds in the fine-tuning process.

## 3 METHOD

LTP comprises two key ideas, *soft-pruning* and *soft $L_0$ regularization*, detailed in sections 3.1 and 3.2, respectively. The full LTP algorithm is then presented in section 3.3.

### 3.1 SOFT PRUNING

The main challenge in learning per-layer thresholds during training is that the pruning operation is not differentiable. More precisely, consider an $N$-layer DNN where the weights for the $l$-th convolutional or fully-connected layer are denoted by $\{w_{kl}\}$, and let $k$ index the weights within the layer. In magnitude-based pruning Han et al. (2015b) the relation between layer $l$'s uncompressed weights and pruned weights is given by:

$$v_{kl} = w_{kl} \times \text{step}(w_{kl}^2 - \tau_l), \qquad (1)$$

where $\tau_l$ denotes the layer's pruning threshold and step(.) denotes the Heaviside step function. We name this scheme hard-pruning. Since the step function is not differentiable, (1) cannot be used to learn thresholds through back-propagation. To get around this problem, during training LTP replaces (1) with soft-pruning

$$v_{kl} \triangleq w_{kl} \cdot \text{sigm}\left(\frac{w_{kl}^2 - \tau_l}{T}\right), \qquad (2)$$

where sigm(.) denotes the sigmoid function and $T$ is a temperature hyper-parameter. As a result of (2) being differentiable, back-propagation can now be applied to learn both the weights and thresholds simultaneously.

Defining soft-pruning as in (2) has another advantage. Note that if $w_{kl}^2$ is much smaller than $\tau_l$ (i.e., $\tau_l - w_{kl}^2 \gg T$), $w_{kl}$'s soft-pruned version is almost zero and it is pruned away, whereas if it is much larger (i.e., $w_{kl}^2 - \tau_l \gg T$), $w_{kl} \approx v_{kl}$. Weights falling within the transitional region of the sigmoid function (i.e., $|w_{kl}^2 - \tau_l| \sim T$), however, may end up being pruned or kept depending on their contribution to optimizing the loss function. If they are important, the weights are pushed above the threshold through minimization of the classification loss. Otherwise, they are pulled below the threshold through regularization. This means that although LTP utilizes pruning thresholds similar to previous methods, it is not entirely a magnitude-based pruning method, as it allows the network to keep important weights that were initially small and removing some of the unimportant weights that were initially large, c.f., Figure 1 (left).

Continuing with equation (2), it follows that

$$\frac{\partial v_{kl}}{\partial \tau_l} = -\frac{1}{2} \cdot \sigma_T(w_{kl}) \quad \text{and} \quad \frac{\partial v_{kl}}{\partial w_{kl}} = \text{sigm}(\frac{w_{kl}^2 - \tau_l}{T}) + w_{kl} \cdot \sigma_T(w_{kl}), \qquad (3)$$

with

$$\sigma_T(w_{kl}) \triangleq \frac{2w_{kl}}{T} \cdot \text{sigm}(\frac{w_{kl}^2 - \tau_l}{T}) \times \left(1 - \text{sigm}(\frac{w_{kl}^2 - \tau_l}{T})\right). \qquad (4)$$

The $\sigma_T(.)$ function also appears in subsequent equations and merits some discussion. First note that $\sigma_T(w_{kl})$ as given by (4) is the derivative of $\text{sigm}((w_{kl}^2 - \tau_l)/T)$ with respect to $w_{kl}$. Since the latter approaches the step function (located at $w_{kl}^2 = \tau_l$) in the limit as $T \to 0$, it follows that the former, i.e., $\sigma_T(w_{kl})$ would approach a Dirac delta function, meaning that its value approaches zero everywhere except over the transitional region where it is inversely proportional to region's width, i.e.,

$$\sigma_T(w_{kl}) \sim \frac{1}{T}, \quad \text{for } |w_{kl}^2 - \tau_l| \sim T. \qquad (5)$$

### 3.2  SOFT $L_0$ REGULARIZATION

In the absence of weight regularization, the per-layer thresholds decrease to zero if initialized otherwise. This is because larger thresholds correspond to pruning more weights away, and unless these weights are completely spurious, their removal causes the classification loss, i.e., $\mathcal{L}$, to increase. Loosely speaking,

$$\frac{\partial \mathcal{L}}{\partial \tau_l} > 0, \text{ unless } \tau_l \text{ is small.}$$

Among the different weight regularization methods, $L_0$-norm regularization, which targets minimization of the number of non-zero weights, i.e.,

$$L_{0,l} \triangleq \sum_k \left| w_{kl} \right|^0,$$

befits pruning applications the most. This is because it directly quantifies the size of memory or FLOPS needed during inference. However, many works use $L_1$ or $L_2$ regularization instead, due to the $L_0$-norm's lack of differentiability. Notably, Han et al. (2015b) utilizes $L_1$ and $L_2$ regularization to push redundant weights below the pruning thresholds.

$L_1$ or $L_2$ regularization methods may work well for pruning older architectures such as AlexNet and VGG. However, they fail to properly regularize weights in networks that utilize batch-normalization layers van Laarhoven (2017), Hoffer et al. (2018). This includes virtually all modern architectures such as ResNet, EfficientNet, MobileNet, and MixNet. This is because all weights in a layer preceding a batch-normalization layer can be re-scaled by an arbitrary factor, without any change in batch-norm outputs. This *uniform* re-scaling prevents $L_1$ or $L_2$ penalties from having their regularizing effect. To fix this issue, van Laarhoven (2017) suggests normalizing the $L_2$-norm of a layer's weight tensor after each update. This, however, is not desirable when learning pruning thresholds as the magnitude of individual weights constantly changes as a result of the normalization. Hoffer et al. (2018), on the other hand, suggests using $L_1$ or $L_\infty$ batch-normalization instead of the standard scheme. This, again, is not desirable as it does not address current architectures. Consequently, in this work, we focus on $L_0$ regularization, which does work well with batch-normalization.

As was the case with hard-pruning in (1), the challenge in using $L_0$ regularization for learning per-layer pruning thresholds is that it is not differentiable, i.e.,

$$L_{0,l} = \sum_k \text{step}(w_{kl}^2 - \tau_l),$$

This motivates our soft $L_0$ norm definition for layer $l$, i.e.,

$$L_{0,l} \triangleq \sum_k \text{sigm}(\frac{w_{kl}^2 - \tau_l}{T}), \tag{6}$$

which is differentiable, and therefore can be used with back-propagation, i.e.,

$$\frac{\partial L_{0,1}}{\partial w_{kl}} = \sigma_T(w_{kl}) \quad \text{and} \quad \frac{\partial L_{0,l}}{\partial \tau_l} = -\frac{1}{2} \sum_k \frac{\sigma_T(w_{kl})}{w_{kl}}, \tag{7}$$

where $\sigma_T(w_{kl})$ is given by (4). Inspecting (7) reveals an important aspect of $L_{0,l}$, namely that only weights falling within the sigmoid transitional region, i.e., $|w_{kl}^2 - \tau_l| \sim T$, may contribute to any change in $L_{0,l}$. This is because other weights are either very small and completely pruned away, or very large and unaffected by pruning. The consequence is that if a significant fraction of these weights, e.g., as a result of back-propagation update, are moved out of the transitional region, $L_{0,l}$ becomes constant and the pruning process stalls. The condition for preventing the premature termination of pruning, when using $L_{0,l}$ can then be expressed as

$$\eta \cdot \left| \frac{\partial \mathcal{L}_T}{\partial w_{kl}} \right| \ll T, \quad \text{for } |w_{kl}^2 - \tau_l| \sim T, \tag{8}$$

where $\mathcal{L}_T$ denotes the overall objective function comprising both classification and soft $L_0$ regularization losses, i.e.,

$$\mathcal{L}_T = \mathcal{L} + \lambda \sum_l L_{0,l}. \tag{9}$$

Note that the left hand side of Eq. (8) is the displacement of $w_{kl}$ as a result of weight update. So (8) states that weight displacement should not be comparable to transition region's width ($\sim T$).

### 3.3 LEARNED THRESHOLD PRUNING

LTP is a magnitude based pruning method that learns per-layer thresholds while training or finetuning. Specifically, LTP adopts a framework of updating all network weights, but only using their *soft-pruned* versions in the forward pass. Gradients for thresholds and weights can be computed using

$$\frac{\partial \mathcal{L}}{\partial \tau_l} = \sum_k \frac{\partial \mathcal{L}}{\partial v_{kl}} \cdot \frac{\partial v_{kl}}{\partial \tau_l} \quad \text{and} \quad \frac{\partial \mathcal{L}}{\partial w_{kl}} = \frac{\partial \mathcal{L}}{\partial v_{kl}} \cdot \frac{\partial v_{kl}}{\partial w_{kl}}, \tag{10}$$

where $v_{kl}$ is the soft-pruned version of the weight $w_{kl}$ as defined by (2). LTP uses (9), (10), (3) and (7) to update the per-layer thresholds:

$$\Delta \tau_l = -\eta_{\tau_l} \left( \frac{\partial \mathcal{L}}{\partial \tau_l} + \lambda \frac{\partial L_{0,l}}{\partial \tau_l} \right). \tag{11}$$

Updating weights needs more care, in particular, minimization of $\mathcal{L}_T$ with respect to $w_{kl}$ is subject to the constraint given by (8). Interestingly, $\partial \mathcal{L}_T / \partial w_{kl}$ as given by (9), (10), (3) and (7), i.e.,

$$\frac{\partial \mathcal{L}_T}{\partial w_{kl}} = \text{sigm}(\frac{w_{kl}^2 - \tau_l}{T}) \cdot \frac{\partial \mathcal{L}}{\partial v_{kl}} + \left( w_{kl} \cdot \frac{\partial \mathcal{L}}{\partial v_{kl}} + \lambda \right) \cdot \sigma_T(w_{kl}), \tag{12}$$

includes $\sigma_T(w_{kl})$ which as a result of (5) could violate (8) for $T \ll 1$ (a requirement for setting $T$, c.f., (15) and Table 1). There are two simple solutions to enforce (8). The first approach is to compute $\partial \mathcal{L}_T / \partial w_{kl}$ as given by (12), but clamping it based on (8). The second, and arguably simpler, approach is to use

$$\frac{\partial \mathcal{L}_T}{\partial w_{kl}} \approx \text{sigm}(\frac{w_{kl}^2 - \tau_l}{T}) \cdot \frac{\partial \mathcal{L}}{\partial v_{kl}}. \tag{13}$$

To appreciate the logic behind this approximation, note that for the vast majority of weights that are outside the sigmoid transitional region, equations (13) and (12) give almost identical values. On the other hand, although values given by (13) and (12), after clamping, do differ for weights within the transitional region, these weights remain there for a very small fraction of the training time (as $\tau_l$ moves past them). This means that they would acquire their correct values through back-propagation once they are out of the transitional region. Also note that (13) is equivalent to only using the classification loss $\mathcal{L}$ (and not $L_{0,l}$) for updating weights, i.e.,

$$\Delta w_{kl} \approx -\eta \frac{\partial \mathcal{L}}{\partial w_{kl}} \quad \text{and} \quad \frac{\partial v_{kl}}{\partial w_{kl}} \approx \text{sigm}(\frac{w_{kl}^2 - \tau_l}{T}), \tag{14}$$

instead of (3), i.e., treating the sigmoid function in (3) as constant. These adjustments are necessary for preventing premature termination of LTP. For example Figure 1 (right) depicts the scatter plot of the pruned model's $w_{kl}^2$ (y-axis) vs. those of the original one (x-axis) for layer3.2.conv2 of ResNet20 on Cifar100 when (3) is used instead of (14). Note how the formation of a gap around the threshold (the red line) causes the pruning process to terminate prematurely with a small threshold. Finally, after training is finished LTP uses the learned thresholds to hard-prune the network, which can further be finetuned, without regularization, for improved performance.

## 4 EXPERIMENTS

### 4.1 CHOICE OF HYPER-PARAMETERS

LTP has three main hyper-parameters, $T$, $\eta_{\tau_l}$, and $\lambda$. Table 1 provides the hyper-parameter values to reproduce the results reported in this paper. LTP uses soft-pruning during training to learn per-layer thresholds, but hard-pruning to finally remove redundant weights. Selecting a small enough $T$ ensures that the performance of soft-pruned and hard-pruned networks are close. Too small a $T$, on the other hand, is undesirable as it makes the transitional region of the sigmoid function too narrow. This could possibly terminate pruning prematurely. To set the per-layer $T_l$ in this paper, the following equation was used:

$$T_l = T_0 \times \sigma_{|w_{kl}|}^2. \tag{15}$$

While one could consider starting with a larger $T_0$ and anneal it during the training, a fixed value of $T_0 = 1\text{e}{-}3$ provided us with good results for all results reported in this paper. One important

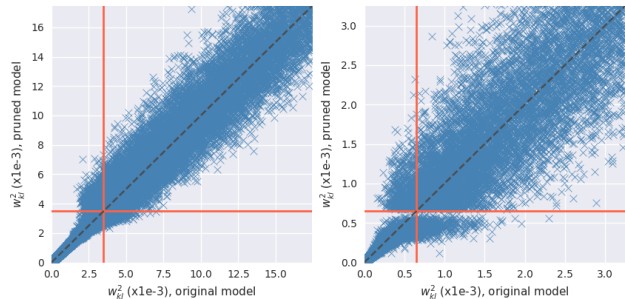

Figure 1: Scatter plot of pruned/original weights for layer3.2.conv2 of ResNet20 (red lines indicate thresholds). Upper-left/lower-right squares show kept/pruned weights that were initially small/large (left). Using (3) instead of (14) stalls pruning by creating a gap around the threshold (right).

| Network | Dataset | $T_0$ | $\eta_{\tau_l}/\eta$ | $\lambda$ |
|---|---|---|---|---|
| ResNet20 | Cifar100 | 1e−3 | 1e−5 | 2e−6 |
| AlexNet | ImageNet | 1e−3 | 1e−7 | 1e−7 |
| ResNet50 | ImageNet | 1e−3 | 1e−7 | 3e−7 |
| MobileNetV2 | ImageNet | 1e−3 | 1e−7 | 1e−7 |
| EfficientNet-B0 | ImageNet | 1e−3 | 5e−7 | 1e−6 |
| MixNet-S | ImageNet | 1e−3 | 1e−5 | 5e−8 |

Table 1: Hyper-parameter values used to produce results reported in this paper.

consideration when choosing $\eta_{\tau_l}/\eta$ is given by equation (10), namely, that the gradient with respect to the pruning threshold gets contribution from the gradients of all weights in the layer. This means that $\partial L/\partial \tau_l$ can potentially be much larger than a typical $\partial L/\partial v_{kl}$, especially if values of $\partial L/\partial v_{kl}$ are correlated. Therefore, to prevent changes in $\tau_l$ that are orders of magnitude larger than changes in $v_{kl}$, $\eta_{\tau_l}/\eta$ should be small. While Table 1 summarizes the values used for $\eta_{\tau_l}/\eta$ for producing results reported in this paper, any value between 1e-5 and 1e-7 would work fine. Finally, $\lambda$ is the primary hyper-parameter determining the sparsity levels achieved by LTP. Our experiments show that to get the best results, $\lambda$ must be large enough such that the desired sparsity is reached sooner rather than later (this is likely due to some networks tendency to overfit, if trained for too long), however too aggressive a $\lambda$ may be disadvantageous as the first pruned model may have poor performance without any subsequent recovery.

## 4.2 ABLATION STUDY

Figure 2 provides an ablation study of LTP with respect to various regularization methods for ResNet20 on Cifar100. As the figure shows, in the absence of regularization, LTP only achieves a modest sparsity of $94\%$ after 100 epochs of pruning (upper-left). $L_2$ regularization achieves good sparsity, but it provides a poor performance (upper-right). This, as explained in section 3.2, is due to the unison and unbound fall of layer weights which deprives $L_2$ loss of its regularizing effect (lower-right). Finally, the keep ratio plot indicates that $L_0$ regularization provides LTP with a natural exponential pruning schedule shown in Zhu & Gupta (2017) to be very effective.

## 4.3 IMAGENET PRUNING RESULTS

In this section we perform an evaluation of LTP on the ImageNet ILSVRC-2015 dataset (Russakovsky et al. (2015)). LTP is used to prune a wide variety of networks comprising AlexNet (Krizhevsky et al. (2017)), ResNet50 (He et al. (2016)), MobileNet-V2 (Sandler et al. (2018)), EfficientNet-B0 (Tan & Le (2019a)) and MixNet-S (Tan & Le (2019b)).

Table 2 gives LTP's ResNet50 pruning results on ImageNet, where it matches the performance achieved by Ye et al. (2019), i.e. $9.11\times$ compression with a Top-5 accuracy of $92.0\%$. LTP also

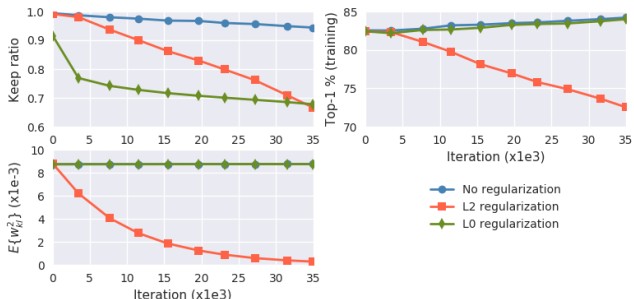

Figure 2: Ablation study with various regularization methods for ResNet20 on Cifar100. Keep-ratio (upper-left), training Top-1 (upper-right) and mean-squared-weights for layer3.2.conv2 (lower-left).

| Method | Top1 | Top-5 | Rate |
|---|---|---|---|
| Original (CaffeNet) | 75.17% | 92.4% | 1× |
| Mao et al. (2017) | N/A | 92.3% | 2.6× |
| Ye et al. (2019) | N/A | 92.0% | 9.16× |
| Original (TorchVision) | 76.15% | 92.9% | 1× |
| Renda et al. | 75.35% | N/A | 9.31× |
| LTP | 73.71% | 92.0% | 9.11× |
| Original (used by STR) | 77.01% | N/A | 1× |
| Kusupati et al. (STR) | 74.01% | N/A | 10.58× |

Table 2: ResNet50 pruning results.

matches Kusupati et al. (2020) performance, considering STR's higher baseline. We note that the iterative weight rewinding method introduced by Renda et al. (2020) provides $1.6\%$ higher top-1 accuracy at a compression rate of $9.31\times$, however it requires 900 epochs of re-training compared to LTP's 30. In fact as Table 3 shows, LTP is very computationally efficient, pruning most networks on ImageNet in less than 100 epochs. This is in sharp contrast to, e.g., Renda et al. (2020), Ye et al. (2019), Han et al. (2015a), He et al. (2018), etc., which typically require a few hundred epochs of training. Finally, Figure 3 provides LTP's top-1 trace (without finetuning) and error-bars across 10 independent runs on ResNet50. As the figure shows, LTP enjoys a low variability in terms of top-1 accuracy of pruned models across different runs.

Table 4 provides LTP's AlexNet performance results on ImageNet, where it achieves a compression rate of $26.4\times$ without any drop in Top-5 accuracy. It is noteworthy that TorchVision's AlexNet implementation is slightly different from CaffeNet's. While both implementations have 56M weights in their fully-connected layers, TorchVision model has only 2.5M weights in its convolutional layers compared to 3.75M of CaffeNet's. As a result of being slimmer, the TorchVision uncompressed model achieves $1.1\%$ lower Top-1 accuracy, and we conjecture can be compressed less.

To the best of our knowledge, it is the first time that (unstructured) pruning results for MobileNetV2, EfficientNet-B0 and MixNet-S are reported, c.f., Table 5. This is partially because LTP, in contrast to

| Method | Scenario | Epochs | Rate |
|---|---|---|---|
| He et al. (2018) | ResNet50 | 376 | 5.13× |
| Renda et al. (2020) | ResNet50 | 900 | 9.31× |
| LTP | ResNet50 | 30 | 9.11× |
| | MobileNetV2 | 101 | 3× |
| | EfficientNet-B0 | 52 | 3× |
| | MixNet-S, | 25 | 2× |

Table 3: Computational complexity results.

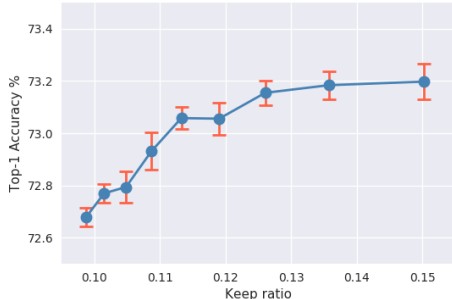

Figure 3: ResNet50 top-1 trace (without finetuning) and error-bars over 10 runs.

| Method | Top-5 | Rate |
|---|---|---|
| Original (CaffeNet) | 80.3% | 1× |
| Han et al. (2015b) | 80.3% | 9× |
| Manessi et al. (2018) | 79.3% | 12× |
| Zhang et al. (2018) | 80.2% | 21× |
| Ye et al. (2019) | 80.2% | 30× |
| Original (TorchVision) | 79.1% | 1× |
| LTP | 79.1% | 26.4× |
| LTP | 78.7% | 29.5× |

Table 4: AlexNet pruning results.

| Method | MobileNetV2 | | | EfficientNet-B0 | | | MixNet-S | | |
|---|---|---|---|---|---|---|---|---|---|
| | Top-1 | Top-5 | Rate | Top-1 | Top-5 | Rate | Top-1 | Top-5 | Rate |
| Uncompressed | 71.8% | 90.4% | 1× | 76.1% | 93.0% | 1× | 76.0% | 92.8% | 1× |
| Global Pruning | 71.2% | 90.1% | 1.33× | 75.7% | 92.7% | 2.22× | 74.5% | 91.6% | 1.67× |
| | 68.1% | 88.3% | 2× | 75.6% | 92.6% | 2.5× | 74.4% | 91.6% | 1.85× |
| | 59.6% | 83.0% | 3× | 75.1% | 92.3% | 3× | 74.5% | 91.6% | 2× |
| LTP | 71.1% | 90.1% | 1.33× | 76.1% | 93.0% | 2.22× | 75.7% | 92.4% | 1.67× |
| | 70.0% | 89.2% | 2× | 75.9% | 92.9% | 2.5× | 75.3% | 92.2% | 1.85× |
| | 68.9% | 88.7% | 3× | 75.2% | 92.4% | 3× | 75.1% | 92.0% | 2× |

Table 5: MobileNetV2, EfficientNet-B0 (without Swish) and MixNet-S pruning results.

many other methods such as, e.g., Han et al. (2015b), Zhang et al. (2018) and Ye et al. (2019), does not require preset per-layer compression rates, which is non-trivial given these networks' large number of layers ($50 \sim 100$), parallel branches and novel architectural building blocks such as squeeze-and-excite. This, along with LTP's computational efficiency and batch-normalization compatibility, enables it to be applied to such diverse architectures out-of-the-box. In the absence of pruning results in the literature, Global-Pruning, as described and implemented in Ortiz et al. (2019), was used to produce baselines. In particular, we see that LTP's $3\times$ compressed MobileNetV2 provides a 9% Top-1 advantage over one compressed by Global-Pruning. Finally, note that LTP can be used to compress MobileNetV2, EfficientNet-B0, and MixNet-S, which are architecturally designed to be efficient, by $1.33\times$, $3\times$ and $2\times$, respectively, with less than 1% drop in Top-1 accuracy.

## 5 CONCLUSION

In this work, we introduced Learned Threshold Pruning (LTP), a novel gradient-based algorithm for unstructured pruning of deep networks. We proposed a framework with *soft $L_0$ regularization* and *soft-pruning* mechanisms to learn the pruning thresholds for each layer in an end-to-end manner. With an extensive set of experiments, we showed that LTP is an out-of-the-box method that achieves remarkable compression rates on traditional (AlexNet, ResNet50) as well as modern (MobileNetV2, EfficientNet, MixNet) architectures. Our experiments also established that LTP gives high compression rates even in the presence of batch normalization layers. LTP achieves 26.4x compression on AlexNet and 9.1x compression on ResNet50 with less than 1% drop in top-5 accuracy on the ImageNet dataset. We are also the first to report compression results on efficient architectures comprised of depth-wise separable convolutions and squeeze-and-excite blocks, e.g. LTP achieves 1.33x, 3x and 2x compression on MobileNetV2, EfficientNet-B0 and MixNet-S respectively with less than 1% drop in top-1 accuracy on ImageNet. Additionally, LTP demonstrates fast convergence characteristics, e.g. it prunes ResNet50 in 18 epochs (plus 12 epochs for finetuning) to a compression factor of 9.1.

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
