# OpenReview forum: "Learned Threshold Pruning"
_ICLR.cc/2021/Conference — Reject_

### Official Review · AnonReviewer3 · 2020-10-13
**Really like the approach, and the exposition; experiments section could be made clearer potentially**

**Rating:** 6
**Confidence:** 3

**Review:**

My overall feeling about this paper is that I really liked it. I felt it was clearly written, nice pacing, didn't bombard us with things we already know, nor did it skip over things we don't know about. I like the approach of using the sigmoids to be able to learn the thresholds and the weight pruning. I really like this approach. In practice, there are still several hyper-parameters to tune, which are apparently model-specific (table 1), and it seems unclear how to directly set the sparsity (we can only influence it using lambda, not set it directly, as far as I can see?). Nevertheless I do really like this approach and direction, ie of learning things as much as possible. The approach seems to me analagous to using eg Adam instead of SGD.

I feel the experiments section could be written a bit more clearly so as to make the assertions in the conclusion and introduction stand out as really obvious. For example, the assertion that LTP can be used in the presence of batch normalization was not made apparent in the experiments section I felt, nor was it shown as a benefit compared to other baselines, I felt.



Details:

'computationally extensive' => 'computationally expensive'

figure 1 really far from where it is first referenced. had to hunt for it....

writing really clear. Good exposition of related papers. Pacing very nice. Very easy to understand.

Good choice of which information to present. (cf papers that present lots of well-known knowledge, or skip over some key advanced concepts).

Kind of a detail, but I like the use of $sigm$ to denote sigmoid, (cf many papers will write the sigmoid out in full, which makes the equations much harder to read)

Personally, I would prefer that $\sigma_T(w_{kl}) = \frac{\partial sigm((w^2_{kl}-\tau_l)/T)}{\partial w_{kl}}$ is defined before equation 3. Otherwise I first read equation 3, wondering what is $\sigma_T$, and then realize it is defined underneath.

To be honest I'm not a fan of the notation $\sigma_T(\cdot)$, since when I first read it I parsed it as $sigm_T(\cdot)$. I would prefer either using some other symbol, or perhaps writing out the full partial derivative, though that seems probably long, without some other symbol, or perhaps adding a $'$, like $\sigma_T'(\cdot)$.

I'd also prefer that the definition of $\sigma_T(w_{kl})$ includes the derivative in this definition, ie:

$$
\sigma_T'(w_{kl}) = \frac{\partial}{\partial w_{kl}} sigm\left(\frac{w^2_{kl} - \tau_l}{T}\right) = ... etc ....
$$

I like the paragraph that describes the behavior of $\sigma_T'$

I like the exposition of the various regularization methods. Concise and yet easy to understand.

$\eta$, in equation 8 is not obviously defined anywhere. I went hunting for it, but couldn't find it. Please define it before using it.

Equation 8 comes out of nowhere, with no explanation of what it means, or why it is. The rest of the paper explains things very well, but equation 8, I'd have to think a lot, wasn't immediately obvious to me where it comes from, what it means.

I'm not sure what the syntax $\sim T$ means here. It normally means 'is distributed as', but that meaning doesn't seem to make sense here? It looks like it's being used to mean $\approx$?.

Ok, I had to go all the way back to equation 2 and the paragraph after it. Looks like $\sim$ is being used here to mean $\approx$. I think that using $\approx$ would be more standard, and easier to understand? Ok, I googled $\sim$, and it turns out that it can often be used to mean 'is approximately the same order of magnitude as', eg https://math.stackexchange.com/a/2177014/45703  But I personally found it confusing because it is very often used to mean 'is distributed as', so personally I would prefer to have a short explanation like 'where $\sim$ means "is approximately the same order of magnitude as"'

I guess the other reason I find it confusing though is this equation implies to me that $|w^2_{kl} - \tau_l| = 0$ would not be in the region, but in fact the region is I feel something like:

$$
|w^2_{kl} - \tau_l| \lesssim T
$$

Personally I think I would prefer the conditions for the transitional region written in this way; would be less confusing for me; I think.

Similarly equation 5 would be:

$$
\sigma_T(w_{kl}) \approx \frac{1}{T}, \text{ for } |w^2_{kl}-\tau_l| \lesssim T
$$

and then equation 8 becomes:

$$
\eta \cdot \left| \frac{\partial \mathcal{L}^T}{\partial w_{kl}}\right| \ll T , \text{ for }  |w^2_{kl}-\tau_l| \lesssim T
$$

I'm not sure though why this derivative is in this constraint? Isnt the constraint simply that

$$
\sum_{l=1}^L \sum_{k=1}^K I\left[ |w^2_{kl} - \tau_l| \right] > m
$$
where $I[\cdot]$ is an indicator function, and $m$ is some positive integer?

ie, at least some points need to be in the transitional region. I'm not sure I follow why we need a derivative in the constraint. Please can you add some description around equation 8 so I can follow what is going on :)

And also trying to work through equation 8, it seems like it is saying that we want to make the derivatives as close to zero as possible, relative to T. But isnt this the opposite of what we want? Dont we want to have a reasonable number of derivatives that are not near zero?

Ok, after equatino 9, we get some explanation for equation 8 :) But I think the explanation could be moved forward somewhat :)))

Ok, based on this explanation, ie the one after equation 9, $\eta$ is probably learnin rate. But please define $\eta$, near equation 8 :)

From the explanation, I'm not sure that equation 8 is a condition that actually *prevents* premature pruning, so much as a heuristic to minimize weights moving too quickly out of the transitional region. Preference to be clearer about this, since it would certainly have helped me to understand equation 8 more easily and quickly :)

equation 11, the brackets could be nicer looking if use "\left(" and "\right)", I feel (so they are as large vertically as the derivative fractions they contain)

$\lambda$ in equation 11 is not defined. From section 4.1, we can see it is a hyper-parameter to be tuned, but that is not stated in equation 11. Preference to state at equation 11 that $\lambda$ is a hyper-parameter, and $\eta_{\tau_l}$ is the learning rate for the threshold of layer l. Hmmm, does this mean that each layer has its own learning rate to tune for its threshold? If there is one single learning rate for the thresholds, I think it might be clearer to represent it as $\eta_\tau$? If there are per-layer learning rates, then this seems to be to contradict the implied promises in the introduction that we don't have per-layer hyper-parameters to set?

Ok, looks like $\lambda$ was first used in equation 9. But still wasnt defined there I think?

I'd also prefer that equation 9 was defined before presenting equation 8, on the whole. This way I can read sequentially, not have to skip forwards and backwards.

The sentence just before and after equation 12 is very complex and hard to take in. Please consider breaking into smaller simpler sentences. eg

"$\partial L_T/\partial w_{kl}$is given by 9, 10, 3 and 7 as:

(equation goes here)

This includes $\sigma_T(w_{kl})$, which from equation (5) $\approx 1/T$, and will become large for $T \ll 1$. This means that the gradient will become large, and constraint (8) will be violated.
"

Figure 1 I feel needs a lot more explanation.
- why are they so symmetrical about the y=x line? doesnt this imply that the number of weights less than theshold before pruning and the number of weights less than theshold after pruning is similar?
- why is the y-axis labeled 'w'? I thought the pruned weights are 'v'?
- why make the plot? What is the motivation of this plot?
- why is the scale of the axes radically different between left and right (0-16 vs 0-3) ?
- why is the left hand plot preferable to the right hand plot?
- why is the gap around threshold in the right hand plot a bad thing?
- why is the proportion of weights below threshold similar in both left and right?


4. experiments

I like the table of hyper-parameters in table 1. (cf many papers skip over which hyper-parameter settings were used, making reproducing the work challenging)

Appreciate the explanation of the significance of $\lambda$ as the primary hyper-paramter determining the sparsity levels. I feel that this explanation could be moved back to equation 9.

Appreciate the observations on how to set $\lambda$, and $\eta_{\tau_l}$.

I wouldnt really call figure 2 an 'ablation study'. It's more like a comparison study of various baselines and approaches I feel? An ablation study I feel would be more like:

- no regularization
- no drop second term in 12 (and simply not use any clamping etc in its place)

I think the ablation study should go after the imagenet pruning results. (I mean, I think it's traditional to put ablation studies after the section of results vs other baselines/sota models)

Table 2 is very unclear to me
- why is your method not at the bottom of the table
- I think your own method should be in bold and say "(ours)" after the name
- looking at the table, it's not very clear why we should choose LTP?
   - the highest rate and top1 looks to be Kusupati et al?
- in the text description, it says that Kusupati uses a stronger baseline, ie STR
   - why don't you use STR too?
- in the text it says that Renda et al needs more training
   - why not put the amount of training required as an additional column in the table?

Jumping to table 3 mid-paragraph is I think jarring. I think first talk about table 2 in one paragrpah, then talk about table 3 in the next. Or at least don't mix and match across tables, at least without having first presented each table on its own first. I feel. Like the sentence 'In fact as table 3 shows' cannot I feel precede the clause 'Finally, figure 3 provides ...', which presents what is table 3. Oh, that's figure 3, not table 3. Anyway, I think table 3 needs some introduction, please.

Yes, so, it seems to me that the resnet50 results from table 3 could be added to table 2 perhaps?

Figure 3 should be in a separate paragraph, since it is not a comparison with baselines/other models. It's just an obseration about LTP itself. And really, without any comparison with how other models/pruning strategies are, I'm not sure it is very meaningful to me? Like, for all I know other models have smaller error bars?

I think table 4 presentation should follow immedaitely table 2 presentation.

Then table 3 presentation.

Figure 3 might be best in a separate 'appendix'-y sub-section at the end of section 4, I feel. Since it's not comparing to other models, like table 2, 4 nad 3 are.

Table 4. If torchvision gives worse results than caffenet, then why not use caffenet instead, or re-implement the caffenet version of alexnet in torch? Otherwise, we can see taht LTP results in table 4 dont match the baselines, and we cannot tell if this is because the torchvision baseline is weaker, and LTP is strong, or whether LTP is weaker than eg Ye et al.

In fact, Ye et al only drops 0.1% top-5 accuracy compared to original, whereas LTP drops 0.4%, compared to torchnet original, so I feel that justifying the lower top-5 error rate on the weaker baseline model is not entirely sufficient?

Table 5 looks like the strongest table to me. Might be worth putting it first? I feel that it could be useful to highlight the top results in each column in each scenario in bold? I think that ideally each column should be a single scenario (whereas here each column is multiple scenarios), then it is easy to highlight the top in each column. For example you could put the different rates as different columns, and use eg top-5 accuracy throughout the table. (or put top 1 and top 5 accuracies as tuples perhaps?)

I kind of think that table 3 should be folded into the other tables.

I think it's not clear from these tables why we should use eg LTP instead of Kusupati et al, or Ye et al. I think that either make it clear in the table somehow, or perhaps put in the text. Like eg "Our method needs considerably less hyper-parameter tuning than existing SoTA methods, whilst achieving nearly the same accuracies for similar levels of compression."

In the conclusion, you mention batch normalization, but this was not brought to the fore in the experiments section. Like, I would expect to see some models that can only be pruned using LTP, and other approaches fail to prune, but I don't remember this being shown clearly in the experiment section?

Basically, I think the assertions in the conclusion are exciting, but aren't made clearly obvious in the experiments section. I think for each assertion in the conclusion there should be a single table or graph that shows this assertion very clearly, in comparison to other possible baselines.

[post discussion edit]

After discussion, I lowered my  score to 'marginally above acceptance threshold':
- the theory section of the paper looks very interesting to me
- I find it hard to see clearly from the experimental section the extent to which the method beats existing methods
- I feel that the experiments could be made more rigorous to clearly show the benefit compared to other techniques
- concretely, I feel taht the tables could be structured in such a way that one can glance at each single table, and see clearly in what way LTP is better than the baselines. Concretely, for the results tables:

- table 2: LTP gives worse accuracy than Renda, and worse compression. The text mentions training epochs are fewer for LTP, but the table doesn't show this benefit (there is no column with number of training epochs)
- table 3: this table is a little apples and oranges I feel. it shows that the number of training epochs is less for LTP than Renda, but the compression ratio is slightly less. I feel that you could have compressed a little more, to make the compression ratios comparable. In addition, I feel it is important to include the accuracy in the table. Without accuracy, then I feel it is not possible to compare.
- table 4: I feel you could do whatever is needed to do to ensure that the baseline model you are using matches the baseline that other teams are using. This could mean porting NTP to caffe, or porting caffe network into torch. Currently, the LTP pruning is on a worse 'parent' model, and performs worse than the other baselines in terms of accuracy. I'm not sure it's sufficient to hand-wavingly just add/subtract the delta in performance between the baselines to the LTP results (which is not explicitly being done, but if one doesn't do that, then one would have to assume that LTP performs worse, I feel)
- which only leaves table 5 that plausibly provides an apples-for-apples comparison, but only for a single baseline

---

> ### Author Response · Authors · 2020-11-20
> **Learned Threshold Pruning**
>
> We thank you very much for your comments. Please find our responses below.
> 1.	We agree that in its current form LTP’s final keep-ratio is only indirectly controlled via $\lambda$. However, an explicit pruning schedule can easily be imposed on LTP where the pruning process is split among several rounds, each targeting a preset keep-ratio. In this form, $\lambda$ is started off with a small value, but gradually increased to make sure each round’s target keep-ratio is reached at the desired point.
>
> 2.	We will make changes to the experiments section to make each assertion supported by either a graph or a table. Regarding efficacy of our soft $L_0$ compared to $L_2$ & $L_1$ in the presence of batch normalization, please refer to our response #11&12 To reviewer #2. For a different perspective (admittedly not related to batch-normalization), please refer to response #2 to reviewer #1.
>
> 3.	We intended to use $\delta(.)$ instead of $\sigma(.)$ as it is often used to denote the Dirac “delta” function. We will fix this and include the derivative in its definition.
>
> 4.	We will use ∼ and ≲, when defining sigmoid function’s transitional region, and clearly state their definitions.
>
> 5.	We will define $\eta$, i.e., learning-rate, before using it. We explained equation (8) after equation (9), but we do agree that this part needs to be revised.
>
> 6.	We will define $\lambda$ after (9), and $\eta_{\tau}$ after (11). There is only one threshold learning-rate, hence we will drop subscript $l$. We will move (9) before (8).
>
> 7.	We will rewrite the section around equation (12) to make it more readable.
>
> 8.	A weight’s distance from y=x indicates how much it changed during pruning. Figure 1 (left) shows that many kept and pruned weights changed only slightly, while some kept weights grow significantly (points to the left and above the red lines) and some pruned weights shrunk a lot (points to the right and below the red curves). The Motivation of this plot was to show that, because of the latter fact, LTP is not strictly a magnitude-based pruning scheme. The motivation of the right plot was to show the importance of (8), which ensures that pruning does not stall.
> The scale of the two plots are different because pruning in the right plot stalled early on, resulting in a much smaller pruning threshold (3.6 vs. 0.7). A gap around the threshold is undesirable as it indicates that sigmoid function’s transitional region is depleted of weights, causing $L_0$ not to vary for small changes in the threshold. This stalls the pruning process (c.f., text between (7) and (8)).
> The scatter plots are not suitable for inferring proportions as a point may represent a single weight, or multiple ones falling on top of one another. We will add more details to Figure 1 to make it clearer.
>
> 9.	Figure 2 compares $L_0$ vs. no regularization (vs. $L_2$). We will move Figure 1, which compares using (3) instead of (14), to the ablation study and move that section after the ImageNet Pruning. This covers both suggested scenarios.
>
> 10.	Table 2 has three sections; each covering comparison points with the same original model (Caffe/TorchVision/STR). We have repeated LTP with STR model and will update Table 2 to include it. This has the added benefit that LTP will be the last entry of TorchVision and STR sections. We will label LTP results as “ours”. For comparison to Kusupati, please refer to our response #7 to reviewer #1. We will add number of epochs to Table 2 to provide a better comparison between LTP and Renda.
>
> 11.	While Figure 3 does not report on other methods, it does provide information on LTP’s consistency (please refer to our response #4 to reviewer #2). We will explain Figure 3 in a separate paragraph and swap presentation of Tables 4 and 3.
>
> 12.	TorchVision’s AlexNet is different from that of Caffe, with the former having fewer parameter in its convolutional layers (2.5M vs. 3.75M). We tried to import Caffe’s pre-trained AlexNet to torch (using https://github.com/jiecaoyu/pytorch_imagenet), but did not succeed. Since the two models, despite differences, bear many similarities, we felt comparing results would still be useful, and we have stated the caveats clearly. We will remove the speculative comment “, and we conjecture can be compressed less” from the revised paper.
>
> 13.	We will present Table 5 first and re-arrange it for enhanced readability.
>
> 14.	With respect to Kusupati, we have repeated LTP on STR baseline and show a more comparable result (please refer to our response #7 to reviewer #1). We note that Ye et al needs 730 epochs to compress AlexNet, whereas LTP needs less than 100 epochs for most networks. We will add the suggested comment “Our method needs considerably less hyper-parameter tuning than existing SoTA methods, whilst achieving nearly the same accuracies for similar levels of compression” to the Table 3’s text.

---

### Official Review · AnonReviewer2 · 2020-10-27
**An exciting new approach to pruning but the execution is flawed**

**Rating:** 4
**Confidence:** 5

**Review:**

## Summary

The paper introduces a new type of soft threshold operator in conjunction with appropriate weight regularization that can be used in the context of neural network pruning to obtain sparse, performant networks from pre-trained, dense networks. The main idea is to replace the Heaviside step function that occurs in "hard threshold" pruning, which is non-differentiable, by a sigmoid function that can be differentiated and thus enables the efficient training/optimization of relevant pruning parameters. Pruning is hereby performed on a per-layer basis by training a regularized per-layer threshold.

## Score

I am quite intrigued by the method and I think it has potential. It seems easy enough to implement while providing decent improvement over competing methods, at least judging from the sparse experimental results that were presented. This brings me to the big weakness and the reason for my score. The experiment section does not provide enough evidence in my opinion to justify acceptance since I cannot say with full confidence that the method reliably performs well. And even when the approach does not outperform existing methods in all aspects, at least I would like to be able to judge what are the scenarios where the method performs well. More details are provided below.

## Ways to Improve My Score

Mainly, please address the points I mentioned in the "Weaknesses". Some of them can be addressed by updating the writing. However, the major concern of mine is the experiment section. I think it requires a full overhaul including more comparison methods, standardized experiment settings, better organized presentation of the results, and clear description of the hyperparameter choices.

## Strengths

* The concept of the introduced soft threshold operator is easy to follow and intuitive. I appreciate the detailed description of the resulting derivatives in Section 3 and the provided intuition. As such the method is well-described and the benefits are clear.

* The various aspects of their ablation studies are interesting, c.f. Figure 1, Figure 2. It helps better understanding some of their design choices.

* The presented experimental evidence seems to hint at very decent performance, especially at the ImageNet scale. This is encouraging to see and underscores the intuition behind the method.

## Weaknesses

* To me, the biggest weakness is the presented experimental evidence. It seems scattered, the presentation is confusing, and most of all it makes it extremely difficult to assess the performance gains of the presented method in comparison to existing methods. Some points that I would like to list specifically are below:
  1. There is no single figure that allows me to assess the prune-accuracy trade-off across a large range of prune ratios and for various comparison methods. Something like Figure 4 in the paper by Kusupati et al. 2020 (https://arxiv.org/pdf/2002.03231.pdf) is, in my opinion, necessary in order to more reliably assess the resulting performance.
  2. The authors only present a selected set of prune ratios and resulting accuracies in their Tables 2-5. Also the results are presented inconsistently. Table 3 only reports compression rate. Table 4 reports Top-5 accuracy and compression rate but not Top-1 accuracy. Table 2, 5 present Top-1 and Top-5 accuracy but only a few selected comparison methods that differ from the one presented in Table 4.
  3. More comparison methods: Unstructured pruning has seen quite a few advancements over the last couple of years and as such I believe it is crucial to compare to many more pruning methods. This is particularly important since standardized comparisons are missing and so simply presenting some results gathered from other papers is not enough in my opinion.
  4. Were experiments repeated multiple times? I can see that Figure 3 was based on 10 repeated runs. What do the error bars represented? Also, what about the other experiments? Could the authors clarify how the other numbers were generated and also report mean and standard deviation. The experiments should be repeated at least a couple of times.

* A more thorough comparison to STR (Kusupati et al. 2020, https://arxiv.org/pdf/2002.03231.pdf). STR shares a lot of similarities with this work in the sense that STR also introduces a per-layer threshold for pruning that can be efficiently optimized using a differentiable soft threshold operator. There is only one comparison point in Table 2, which however seems to be based on a different implementation thus resulting in a different baseline accuracy. There is also no discussion how the soft threshold operators of STR and LTP differ and what makes one better than the other.

* The experimental hyperparameters are not fully listed and the ones that are listed are scattered throughout the paper. Also, the authors did not provide code; so I couldn't check their implementation either. In particular:
  1. The contributions list in the introduction mentions the number of pruning and fine-tuning epochs for some experiments but the experiment section doesn't provide a full overview of pruning+fine-tuning epochs. Table 3 provides some of these numbers but not all details. Also what are the particular reasons for the choices? ResNet50 seems to require 30 total pruning epochs, while MobileNetV2 requires 101 epochs. Why?
  2. What does $\sigma^2_{|w_{kl}|}$ in equation 15 refer to?
  3. What about training parameters? Are those the same as in the original paper? What about an overview table with all training parameters?
  4. How were the comparison methods implemented? Were they even implemented or were the results taken from the respective papers?

* I am not convinced that $L_1$ and $L_2$ regularization don't work for pruning in general as the authors claim in Sections 2 and 3.2. I understand their point and in their case their approximation to $L_0$-regularization indeed seems to play a central role but I wonder how true this statement is in general. In particular, while batch normalization (BN) may allow for arbitrary re-scaling of layers, the _relative magnitude_ of weights may still be impacted by $L_1$-regularization. Moreover, after all they don't use $L_0$-regularization either, just another differentiable approximation to $L_0$-regularization (just like $L_1$ is a differentiable approximation to $L_0$).

## Other Minor Feedback

* I find the introduction and related work interesting and it serves as an appropriate motivation for the work. However, I find it somewhat disingenuous and/or misleading not to mention global magnitude-based pruning in the first sections. The authors cite a lot of related work that requires manual and/or more complicated approaches to identifying per-layer sparsity patterns when the most obvious solution is to perform global magnitude pruning (GMP). Since this usually works quite well as baseline and the authors also compare to GMP in their experiment section, I believe this merits a longer discussion where the authors compare to GMP.

* Section 3.3 could be simplified. I was pretty confused since a lot of prior quations are cited in the explanation between newly introduced equations. That makes it pretty hard to read and I believe the introduced concepts and resulting equations could be streamlined. E.g. you could introduce all relevant equations in a continuous paragraph instead of jumping between equations, which results in the heavy use of equations citations.

---

> ### Author Response · Authors · 2020-11-20
> **Learned Threshold Pruning**
>
> We thank you very much for your comments. Please find our responses below.
> 1.	We appreciate reviewer’s desire for an overhaul of our results, where SOTA methods are implemented & top1 plots for a range of keep-ratios are given, but we hope following facts demonstrate that our method of identifying relevant SOTA & comparing against them is both effective & practical. First, reproducing SOTA results is non-trivial (Zhang reported 21x compression for AlexNet but using their code we could not go above 10x). Even if code/hyperparameters are given, producing so many points, each requiring 100’s of epochs, is computationally very intensive. Second, we note that GMP uniformly beats all other baselines in Figure 4 of Kusupati and that Renda, Zhang and Ye (all beating GMP) are absent.
> 2.	Many SOTA works (Mao, Zhang, Ye) only present a few comparison points as a table (Kusupati also gives MobileNetV1 results as a table). Tables are effective as they focus on inflection points of tradeoff curves where accuracies are still high & useful; reporting very high or low keep-ratios where accuracy has not dropped at all, or dropped too much, is not useful.
> Table 3 compares computational efficiency of methods; hence only reports number of epochs needed to achieve a certain compression rate. Tables 2 and 4 report Top1 and Top5 wherever the original paper provided them. Tables 2, 4 and 5 pertain to different DNN’s, with differing baselines as each paper only considered a subset of them. We will add comments to clarify these points.
> 4.	We agree that unstructured pruning has seen many advances & provide a thorough account in our Related Work. Cognizant of this, we identified SOTA works for each DNN, e.g. Mao, Ye, Renda, Kusupati for ResNet50, and Han, Manessi, Zhang, Ye for AlexNet, and compared to them. For MobileNetV2, EfficientNet-B0 & MixNet-S there are no results in the literature yet, hence we used GMP as baseline. We will augment Table 2 with more comparison methods, e.g., GMP, DNW, etc.
> 5.	Figure 3 shows error bars for 10 runs of LTP on ResNet50. The bars give top1s’ standard deviation, centered at average top1 for these 10 runs. The plot attests to LTP’s consistency; typical standard deviation is around 0.1%. We will clarify this in Figure 3.
> 6.	For performance comparisons to STR, c.f., response #6&7 to reviewer #1. While STR and LTP’s soft pruners both allow for learning per-layer thresholds, latter also allows for keeping small but important weights & pruning large but redundant ones, which differentiates LTP from strictly magnitude-based methods (c.f., responses #1 and #2 to reviewer #1). LTP’s soft pruner also provides a parameter (Sigmoid’s $T$) as a control knob. Kusupati does not comment on whether theirs allows for such behavior. We will add comments to Figure 1 to further clarify.
> 7.	For hyperparameters, please refer to our response #5 to reviewer #1.
> 8.	We will add number of pruning and finetuning epochs to Table 3. LTP does not have an explicit pruning schedule, instead it exhibits a natural polynomial-like schedule as seen in upper-left plot of Figure 2 (Zhu and Gupta 2017 also reported that a polynomial schedule gave them their best results). $\lambda$ determines how rapidly LTP gets to target keep-ratio. MobileNetV2 is architecturally more compact than the over-parameterized ResNet50, therefore we set $\lambda$ such that LTP pruned it more gently, which translated to a larger number of epochs. We will add comments to Table 3 clarifying this.
> 9.	$\sigma_{kl}^2$ is the variance of layer $l$’s empirical weight distribution.
> 10.	LTP operates on pre-trained models, hence oblivious to training parameters of original papers. We used a learning rate of 1e-3, a batch size of 128 and the SGD optimizer (momentum of 0.9) for all results. We will add these details to hyperparameter section.
> 11.	Except for Table 5, all points are from original papers.
> 12.	Deficiency of $L_1$ and $L_2$ has been discussed in Laarhoven 2017 & Hoffer 2018, & ML blogs such as https://blog.janestreet.com/l2-regularization-and-batch-norm/. We have provided evidence in our ablation study. Top two plots of Figure 2 show that LTP with $L_0$ or $L_2$ prunes ResNet20 to a keep ratio of 0.7 in 35 epochs. With $L_0$ the model keeps its original training top1 of 85%, whereas $L_2$ causes it to drop to 73%. Lower left plot shows that magnitude of kept weights remains constant under $L_0$ but exponentially drops with $L_2$, in accordance with section 3.2.
> 13.	We agree that $L_1$ provides some regularization, but as observed, only to the extent that it approximates $L_0$. Equation (6) reveals that our soft $L_0$ is in fact an $L_1$ norm that is applied to thresholded, rather than the original, weights. Hence it is a better approximation to $L_0$ than $L_1$ (see also response #2 to reviewer #1). We will add comments to further clarify this.
> 14.	We will add more details on GMP in our Related Work & revise section 3.3 to streamline it more.

---

### Official Review · AnonReviewer4 · 2020-10-28
**Review of the paper Learned Threshold Pruning**

**Rating:** 6
**Confidence:** 4

**Review:**

The paper proposed a new method to prune a neural network. The method is interesting, innovative and effective. It makes it possible to learn tunning parameter via back propagation, hence learn together with network's weights.
The work is well motivated.
The paper is well structured, the writing is clear and easy to follow.
The conducted experiments are thorough and clearly show the efficiency of the proposed method. The paper contains enough information to replicate the experiments.

The work would be beneficial for others if the code is published open.

A question for clarification: When hard prunning the network (section 4.1), we just replace sigmoid(x) by step(x)?

Post-discussion update: I have read the updated paper and other reviews, especially from reviewer #2. While I am still positive about the approach/methodology, I am not confident about the technical details of the experiments, without which, it's very hard to justify the effectiveness of the method. I share other reviewers' views regarding inconsistencies, e.g. Tables 2-5, that have not been fixed in the updated paper.

---

> ### Author Response · Authors · 2020-11-20
> **Learned Threshold Pruning**
>
> We thank you very much for your comments.
>
> As the reviewer notes, LTP replaces the sigmoid function by the step function to hard prune the network at the end of the pruning process. As stated in section 4.1, selecting a small enough T (e.g., 1e-3 times the variance of layer’s empirical weight distribution, c.f., equation (15) and Table 1) ensures that the performance of the soft-pruned and hard-pruned networks are close, and resulting performance loss, if any, would be recovered during the subsequent finetuning of the hard pruned network.

---

### Official Review · AnonReviewer1 · 2020-10-28
**A parameter based pruning method based on a relaxation of L0 regularization and a novel per layer threshold learning**

**Rating:** 4
**Confidence:** 5

**Review:**

This paper proposes soft pruning and soft l0 regularization.

The soft pruning learns a binarization function based on a sigmoid. I am curious to understand how the algorithm can let some low magnitude weights not be pruned and some large magnitude weights be pruned in the end. The paper claims the key is updating all the weights while maintaining the masks for inference. That seems to me ineffective compared to L1 or similar methods. Also, even in the case the weights are not masked, eventually, those gradients should become zero (as they are masked in the forward pass) and therefore preventing them from having large values, isn't it? That needs some clarification.

As the paper then focuses on L0 regularization, I missed comparisons to related work on that regard. This is not the first paper aiming at using L0 and proposing a differentiable approach. How this compares to others?  For instance, a quick search led to LEARNING SPARSE NEURAL NETWORKS THROUGH L0 REGULARIZATION in ICLR 2018.



I am confused with one of the contributions is to "provides a trace of checkpoints with varying pruning ratios and
accuracies. Because of this, the user can choose any desired checkpoint based on the sparsity
and performance requirements for the desired application"
Why is this different from any other approach? As soon as the code saves the checkpoint (which most do) then, the user has access to the same flexibility, right?


The section about the hyperparameters is confusing. How are the hyperparameters determined? Each architecture is using a different hyperparameter, how a user could set these?

The experimental results do not seem to support the novelty and the text is kind of misleading. LTP is below Renda and Kusupati. Text suggests Kusupati uses a better baseline, would it be possible to show results of LTP on that baseline? if not, why not?
For Renda, seems like the key difference is the training process. Would be good to see the benefits of a longer training process for LTP. It is not clear to me that LTP can get better results even training for longer.

Results on more compact networks compared to the self-implementation of Global pruning seem promising for larger compression rates.

The paper also claims the benefit of learning the threshold per layer, however, provides no result on the distribution of those parameters. Would be interesting to see how these values are distributed in each architecture to reinforce the value of not using a single value for all layers and architectures.

---

> ### Author Response · Authors · 2020-11-20
> **Learned Threshold Pruning**
>
> We thank you very much for your comments. Please find our responses below.
> 1.	LTP keeps a small but important weight, $w_{kl}$ by constantly pushing it above the threshold; assume that the threshold grows large enough such that $w_{kl}$’s mask drops just below $1$, e.g., $0.95$, resulting the soft pruned weight $v_{kl}$ to drop from its locally-optimal value of $w_{kl}$ to $0.95 \times w_{kl}$. Since $v_{kl}$ is important, the backprop restores it to its original value by scaling up $w_{kl}$ by a factor of $1.05 = 1/0.95$. Note that this process of pushing small but important weights above the threshold is due to LTP’s soft-pruning, not soft regularization ($L_0$ or otherwise). We will add clarifying details to Figure 1.
> 2.	LTP prunes a large but redundant weight, $w_{kl}$, by dragging it below the threshold. Assume the threshold grows large enough such that $w_{kl}$ just enters sigmoid’s transitional region. As such, $w_{kl}$ receives a second gradient with respect to $L_0$, in addition to gradient with respect to classification loss. Since $w_{kl}$ is redundant, $L_0$ gradient is dominant and causes $w_{kl}$ to decay below threshold. While $L_2$ or $L_1$ may have a similar effect, using $L_0$ is advantageous as it targets weights next to pruning thresholds (within the transitional region), whereas $L_2$ or $L_1$ place more importance on larger weights that are further away. This allows $L_0$ to better measures various layers pruning potential. Also, $L_0$ is not affected by batch normalization as detailed in the paper. We will add clarifying details to Figure 1.
> 3.	We have provided a thorough review of related works in our paper. The referenced work, (Louizos et al 2018), has been cited in the first paragraph of Section 2. Also, (Louizos et al 2018) provided some results for MNIST and CIFAR, but none for ImageNet, which is the focus of our paper.
> 4.	Most pruning methods, iterative magnitude-based pruning (Han et al 2015a and b), weight & learning-rate rewinding (Renda et al), progressive ADMM (Ye et al 2019) operate according to a pruning schedule where pruning is split among several rounds, each targeting a specific pruning ratio subject to a computational budget (~40 epochs for ImageNet). As such, they do not provide a continuum of pruning ratios; if the accuracy at the end of a pruning round is not acceptable, one needs to repeat the last few pruning rounds (potentially involving tens of epochs) with a less aggressive schedule. This contrasts with LTP where toward the end, the keep ratio and model accuracy change very gently from one epoch to the next. Hence in a single run of LTP, we get a series of checkpoints with different pruning ratio vs accuracy tradeoffs to choose from.
> 5.	LTP has 3 hyper parameters: $L_0$ loss multiplier ($\lambda$), sigmoid temperature multiplier ($T_0$) and threshold learning-rate ratio ($\eta_{\tau} / \eta$). $\lambda$ controls the final pruning ratio and is the only parameter that needs per scenario adjustment. The $\lambda$ values in Table 1 provide guidance on how to choose for a new architecture. For all results in the paper we have used a fixed $T_0$ of 1e-3 and although we have used 3 values for $\eta_{\tau} / \eta$, our hyper-parameter search showed that any value between $1e-5$ and $1e-7$ works as fine. Hence $T_0 = 1e-3$ and $\eta_{\tau} / \eta  = 1e-5$ provide a reasonable choice for a new architecture. LTP is very robust to hyper-parameters. We will add some clarifying comments to the hyperparameter section.
> 6.	Regarding novelty, LTP and Kusupati (a contemporaneous work as they cite LTP), are among the first differentiable methods for unstructured pruning of neural networks. Also, LTP’s differentiable $L_0$ regularizer is quite novel. In addition, we are the first to report unstructured pruning results (beyond global magnitude-based pruning) for MobileNetV2, EfficientNet-B0 and MixNet-S. We will add comments to better clarify novel aspects of LTP.
> 7.	Regarding Kusupati, we had used TorchVision’s pretrained model (a standard model) as baseline. Kusupati used another model with a 0.85% top1 accuracy advantage. Based on reviewer’s suggestion, we have repeated LTP with Kusupati’s baseline and achieved a top1-accuracy of $74.38%$ at a compression rate of 8.84x (keep percentage of 11.3%) after 28 epochs, compared to Kusupati’s top1 of $74.31$ at a compression rate of 10.24x (keep percentage of 9.8%). We will update Table 2 to include this comparison point & change “matching“ to “comparable” in its text.
> 8.  Renda requires 900 epochs compared to LTP’s 30, a significant difference. We agree that weight and learning-rate rewinding are retraining and not pruning schemes (Renda uses global threshold pruning). Hence, weight or learning-rate rewinding can easily be used on top of LTP and likely improve its performance. We will test this in our future work.
> 9.  We will add a new figure showing per-layer keep-ratios & pruning thresholds.

---

### Decision · Program_Chairs · 2021-01-07
**Final Decision**

**Decision:**

Reject

**Comment:**

This paper proposes a method for differentiable pruning that replaces the hard thresholding of standard pruning, with a soft version that permits taking the gradients of the pruning threshold. The proposed benefits are an accuracy that is better or competitive with alternative methods as well. Moreover, the paper suggests the technique to be efficient.

The pros of this paper are that it is working in an interesting setting of differentiable pruning, with the hope of -- in some sense -- simplifying the pruning process or at least unifying the process with standard training.  The technique is plausibly justified in its technical development. The paper also follows with a significant number of experiments.

The cons of this paper are that the conceptual framework -- beyond the initial idea -- is not fully clear. In particular, this paper does not elucidate a clear set of claims and hence, results in the difficulty on the Reviewers part in detangling the claims and identifying the appropriate comparisons.

For example, the paper doesn't take up a simple claim that it is state-of-the-art in accuracy vs parameter measures (and would seem not to given the results of Renda et al. (2020)).  It need not necessarily make this claim, but there are suggestions to such a claim early in the paper. If this is not an intended claim, then the paper can remove any suggestions to such (i.e., the claims around new SoTA for networks not evaluated in prior work).

The paper has a somewhat tentative claim that it is more efficient (in the total number of epochs of training) versus other techniques (Table 3).  However, the presented results are only at a single-point versus other methods.  Renda et al. (2020) directly consider accuracy versus retraining cost trade-offs. Appendix E of that paper provides one-shot pruning results for ResNet-50 showing accuracy on par with that presented here.  The number of retraining epochs is also similar to here. This paper, however, only compares against the most expensive iterative pruning data point in the other paper.

In sum, my recommendation is Reject. This is promising work that needs only (1) to include a few testable claims and (2) to re-organize the results (and perhaps run a limited set of new results) to thoroughly explore those claims. For example, if the most important claim is accuracy vs retraining cost, then it needs to show a more complete trade-off curve of the two results.  Of course, this, in principle, opens the door to comparisons to many other techniques in the literature.